# Adhesion Forces of Oral Bacteria to Titanium and the Correlation with Biophysical Cellular Characteristics

**DOI:** 10.3390/bioengineering9100567

**Published:** 2022-10-17

**Authors:** Katharina Doll-Nikutta, Andreas Winkel, Ines Yang, Anna Josefine Grote, Nils Meier, Mosaieb Habib, Henning Menzel, Peter Behrens, Meike Stiesch

**Affiliations:** 1Department of Prosthetic Dentistry and Biomedical Materials Science, Hannover Medical School, Carl-Neuberg-Str. 1, 30625 Hannover, Germany; 2Lower Saxony Centre for Biomedical Engineering, Implant Research and Development (NIFE), Stadtfelddamm 34, 30625 Hannover, Germany; 3Institute for Technical Chemistry, Technische Universität Braunschweig, Hagenring 30, 38106 Braunschweig, Germany; 4Institute of Inorganic Chemistry, Leibniz University Hannover, Callinstr. 9, 30167 Hannover, Germany; 5Cluster of Excellence Hearing4All, 30625 Hannover, Germany

**Keywords:** atomic force microscopy, single-cell spectroscopy, bacterial adhesion, cell surface, cell respiration, dental implant

## Abstract

Bacterial adhesion to dental implants is the onset for the development of pathological biofilms. Reliable characterization of this initial process is the basis towards the development of anti-biofilm strategies. In the present study, single-cell force spectroscopy (SCFS), by means of an atomic force microscope connected to a microfluidic pressure control system (FluidFM), was used to comparably measure adhesion forces of different oral bacteria within a similar experimental setup to the common implant material titanium. The bacteria selected belong to different ecological niches in oral biofilms: the commensal pioneers *Streptococcus oralis* and *Actinomyces naeslundii*; secondary colonizer *Veillonella dispar*; and the late colonizing pathogens *Porphyromonas gingivalis* as well as fimbriated and non-fimbriated *Aggregatibacter actinomycetemcomitans*. The results showed highest values for early colonizing pioneer species, strengthening the link between adhesion forces and bacteria’s role in oral biofilm development. Additionally, the correlation between biophysical cellular characteristics and SCFS results across species was analyzed. Here, distinct correlations between electrostatically driven maximum adhesion force, bacterial surface elasticity and surface charge as well as single-molecule attachment points, stretching capability and metabolic activity, could be identified. Therefore, this study provides a step towards the detailed understanding of oral bacteria initial adhesion and could support the development of infection-resistant implant materials in future.

## 1. Introduction

Bacterial adhesion to surfaces is a widespread phenomenon, as it only requires the respective microorganisms, an interface to adhere to and nutrient supply. In dentistry, this process is responsible for the development of illnesses, such as periodontitis at the tooth and peri-implantitis at dental implants. The reason for this is that bacterial adhesion initiates the formation of biofilms—complex three-dimensional multispecies agglomerates that are surrounded by a self-produced matrix of extracellular polymeric substances and in which the bacteria exhibit a unique phenotype [1,2]. Biofilms cause inflammatory tissue reactions and show an inherent resistance towards antibiotic therapy [3,4]. Oral biofilms are formed by a multitude of different bacterial strains that adhere to the solid tooth or implant surface, as well as to each other in a specific species order [5]. They can be divided into commensal early colonizers that are associated with oral health, secondary or bridge bacteria and late colonizing oral pathogens that form distinct ecological niches within the biofilm [6]. Characterizing the interaction of these different species with implant materials sets the basis towards the development of anti-biofilm strategies.

Bacterial cell force spectroscopy based on atomic force microscopy (AFM) is as versatile technique to analyze bacteria–surface interaction by directly measuring bacterial adhesion forces. Bacterial cells are immobilized on an AFM cantilever tip of known spring constant and sensitivity. Then, the cantilever-attached bacteria are brought into contact with the surface of interest and deflection of both the approach and the withdraw process is monitored as a function of the tip-surface distance. Changes in bacterial adhesion forces at nN and µm scales can be analyzed as specific deflections in the force–distance curve. As individual bacterial surface sensing leads to heterogeneous microenvironments in bacterial biofilms, measuring bacterial adhesion forces on a single-cell level is advantageous [7]. The critical step for this method, single-cell force spectroscopy (SCFS), is preparing the bacteria-loaded cantilever. For this purpose, chemical fixation, gluing or drying were often applied to coat bacteria irreversibly on the cantilever [8]. In a more recent system, the AFM is connected to a microfluidic pressure control system (FluidFM, Cytosurge AG, Zurich, Switzerland) [9,10]. By using hollow cantilevers carrying a pyramidal, open tip, single bacterial cells can be immobilized non-invasively by applying negative pressure, measured and released by positive pressure, which allows the cantilever to be re-used for further bacterial cells [11]. A previous study could show that FluidFM SCFS results are comparable to SCFS with coated cantilevers [12]. Bacterial force spectroscopy by both systems has already been used to analyze adhesion forces of several oral bacteria, mainly with a focus on viridans group streptococci [13,14,15,16,17,18]. However, to the best of our knowledge, no study has analyzed several species belonging to different niches in oral biofilms within one experimental setup so far. This would allow for direct comparison without perturbation by external conditions. 

It is well known that bacterial adhesion to solid surfaces, such as implant materials, is influenced by a multitude of different (bio)physical factors. Regarding the material surface, these include, roughness, wettability and surface charge [19,20]. Further, biophysical cellular properties contribute to bacterial adhesion: the influence of cell surface charge as well as hydrophobicity and, thus, in a broader sense, membrane integrity has been demonstrated [19,21]; metabolic activity leads to the production of cell surface adhesins that guide initial bacterial attachment; and differences in membrane rigidity (cell elasticity) were shown to be responsible for species-dependent attachment behavior [19,22,23]. However, in contrast to material properties, a systematic investigation how biophysical cellular properties correlate with bacterial adhesion forces has not yet been conducted. In regard to the diverse oral microbiome, it would be especially interesting to see whether there is a general correlation across different species. 

Therefore, the aim of the present study was to step towards the characterization of oral bacteria adhesion to implant materials by (i) comparatively measuring adhesion forces of representatives of different ecological niches in oral biofilms using SCFS and (ii) evaluating the correlation between biophysical cellular parameters and bacterial adhesion across species. The strains selected were commensal pioneer bacteria (*Streptococcus oralis* and *Actinomyces naeslundii*), a secondary colonizer (*Veillonella dispar*) and late colonizing pathogens (*Porphyromonas gingivalis* and *Aggregatibacter actinomycetemcomitans*). They also exhibit different cell wall types (Gram positive and Gram negative). Additionally, for *A. actinomycetemcomitans* (*A. ac*), two subtypes, which differ in their surface molecules (referred to as rough and smooth), were investigated. SCFS was performed using the FluidFM system with two buffers as simplified different environmental conditions. For biophysical cellular parameters, cell elasticity, bacterial surface charge, membrane integrity and metabolic activity were quantified and differences between both buffers were correlated to respective values of adhesion force measurement by linear regression.

## 2. Materials and Methods

### 2.1. Titanium Specimen Characterization

As test specimens, titanium grade 4 discs with 12 mm in diameter were used. They were finished with 45 µm diamond polishing wheels. Surface roughness was determined qualitatively by reflection microscopy (excitation 405 nm, emission 400–410 nm; Leica TCS SP8, Leica Microsystems, Mannheim, Germany) and quantitatively by profile method according to DIN EN ISO 3274 using a tactile surface measuring device (Marsurf M400, Mahr GmbH, Göttingen, Germany). Water contact angle was measured using an optical contact angle measuring device (OCA 40, Software SCA 202 V.3.61.4, DataPhysics Instruments GmbH, Filderstadt, Germany) with a water droplet of 20 µL. Titanium surface charge was determined from specimens of equal material properties and roughness, but of 1 × 2 cm in size in two different buffer systems: phosphate-buffered saline (PBS, Biochrome GmbH, Berlin Germany; containing 136.9 mM NaCl, 2.7 mM KCl, 8.7 mM Na_2_HPO_4_, 2.0 mM KH_2_PO_4_, pH 7.4, ionic strength 166.4 mmol/L) and anaerobe-reduced transport fluid (RTF; containing 2.6 mM K_2_HPO_4_, 15.4 mM NaCl, 6.8 mM (NH_4_)_2_SO_4_, 3.3 mM KH_2_PO_4_, 0.8 mM MgSO_4_·7xH_2_O, 3.8 mM Na_2_CO_3_, 1.0 mM Na_4_EDTA·2xH_2_O, 1.3 mM dithiothreitol, pH 7.0, ionic strength 63.2 mmol/L) [24]. The specimens were mounted in an adjustable gap cell and investigated by streaming current measurement (SurPASS 3 Analyser, Anton Paar GmbH, Graz, Austria). PBS and RTF buffer were diluted with ultrapure water to match the conductivity of the KCl reference solution (1 mM, 15 mS/cm ± 1 mS/cm). Zeta potential was examined over a pH range from 2.0 to 10.0 using 50 mM HCl and KOH for auto titration by the device. A pressure difference from 200 to 800 mbar was applied to generate the streaming, whereby the linear region from 200 to 550 mbar was used for zeta potential calculation. In some cases, the pressure difference for the calculation had to be reduced to ensure linearity. The zeta potential (ζ) was calculated with the Helmholtz–Smoluchowski equation:(1)$$\zeta =\frac{{\mathrm{dI}}_{s}}{\mathrm{dp}}\times \frac{\eta }{{\varepsilon \times \varepsilon }_{0}}\times \frac{L}{A}$$

Hereby, ζ is the zeta potential [V], dI_S_ is the streaming current [A] and dp the differential pressure [Pa], η is the electrolyte viscosity [Pas], ε is the dielectric coefficient, ε_0_ is the permittivity [A·s·V/m] and L/A is the cell constant [cm^−1^].

### 2.2. Bacterial Strains and Culture Conditions

*Streptococcus oralis* ATCC^®^ 9811 was obtained from the American Type Culture Collection (ATCC^®^, Manassas, VA, USA). The strain was stored at −80 °C as glycerol stocks and routinely pre-cultured in Todd-Hewitt Broth (Oxoid Limited, Hampshire, UK) supplemented with 10% yeast extract (THBy, Carl Roth GmbH + Co. KG, Karlsruhe, Germany) for 18 h under agitation at aerobic conditions and 37 °C. *Actinomyces naeslundii* DSM 43013, *Veillonella dispar* DSM 20735 and *Porphyromonas gingivalis* DSM 20709 were obtained from the German Collection of Microorganisms and Cell Cultures GmbH (Braunschweig, Germany). These strains were stored at −80 °C as glycerol stocks and routinely pre-cultured in Brain–Heart Infusion (BHI, Oxoid Limited) supplemented with 10 µg/mL vitamin K (Oxoid Limited) for 24 h under static, anaerobic conditions and 37 °C. *Aggregatibacter actinomycetemcomitans* JP2 strain (HK1651, CCUG 56173) was obtained from the Culture Collection of the University of Gothenburg (Gothenburg, Sweden). Rough and smooth colonies were isolated from streak plates (THBy + 5% sheep blood + 12 g/L agar agar (both Oxoid Limited)) and separately stored at −80 °C as glycerol stocks, as described previously [25]. Prior to experiments, the strains were pre-cultured in THBy for 72 h under static conditions in 5% CO_2_ and 37 °C. The basic characteristics of the bacterial species regarding shape, size, cell wall structure and adhesion molecules are listed in Table 1.

### 2.3. Bacterial Single-Cell Force Spectroscopy (SCFS)

Bacterial pre-cultures were diluted in filtered PBS or RTF to an optical density of 0.005 at 600 nm. For bacterial SCFS, a FlexFPM atomic force microscope (Nanosurf AG, Liestal, Switzerland) was connected to a microfluidic FluidFM pressure control system (Cytosurge AG) and mounted on an inverse microscope (Eclipse Ti-S, Nikon GmbH, Düsseldorf, Germany). The system was equipped with hollow silicon nitride cantilevers with a circular opening of 300 nm at the end of a pyramidal tip and a theoretical spring constant of 0.6 N/m (FluidFM Nanopipette, Cytosurge AG). The exact spring constant of each cantilever was determined by thermal tuning using software implemented scripts. Values ranged from 0.45 to 0.75 N/m. Cantilevers were filled with degassed, filtered PBS, connected to the microfluidic pressure control system and the sensitivity was calibrated using software implemented scripts [30]. For force spectroscopy, glass dishes (WillCo Wells B.V., Amsterdam, The Netherlands) were equipped with a glass ring into which the disc shaped titanium specimens were inserted at grade. These dishes were filled with the prepared bacterial suspension. To prepare the bacteria-loaded cantilever, a single bacterial cell sedimented on the glass ring was targeted microscopically. The cantilever was brought in contact with this bacterial cell using a setpoint force of 10 nN and a negative pressure of 400 mbar for 5 s. The captured bacterium was retracted with a velocity of 1 µm/s and transferred to the (non-transparent) titanium surface. SCFS was performed with 15 individual cells per bacterial strain in both buffer systems. For each cell, 16 force spectroscopy curves at different positions with a setpoint force of 0.75 nN, an adhesion time of 5 s, a velocity of 1 µm/s and force feedback enabled were recorded. The resulting force–distance curves were analyzed using the AtomicJ 1.7.2 software [31] after passing quality control (remove curves with artefacts due to interference on the rough sample). The maximum adhesion force, the number of attachment points and the detachment distance were calculated from the withdraw curve, as illustrated in Figure 1. The Young’s modulus, as measure of cell elasticity, was fitted from the approach curve. All analysis settings are specified in Appendix A.

### 2.4. Bacterial Zeta Potential Measurement

Bacterial zeta potential was determined by diffusion barrier method. Bacteria were prepared as described above and diluted in either PBS or RTF to a final optical density of 0.005 at 600 nm. For this purpose, first buffer was injected into a folded capillary cell (Malvern Panalytical GmbH, Nürnberg, Germany). Then, bacteria were added by direct application at the lowest point of the capillary cell using a syringe. Measurements were carried out at 20 V and 100 V with maximum measurements of 20 and 40 for PBS and RTF, respectively, using the Zetasizer Nano ZSP (Malvern Panalytical GmbH).

### 2.5. Fluorescence Staining and Confocal Laser-Scanning Microscopy

To analyze bacterial membrane integrity, bacterial cultures were prepared as described above, added to the experimental setup for SCFS (dish with glass ring and titanium) and incubated for 5 h under ambient conditions. This resembled the maximum time needed for SCFS. Following incubation, the sedimented bacterial cells were stained using the LIVE/DEAD BacLight Bacterial Viability Kit (Life Technologies, Darmstadt, Germany). The two fluorescent dyes, Syto^®^9 and propidium iodide, were applied simultaneously at 1:2000 dilution in PBS or RTF, respectively, according to the manufacturer’s instructions. Bacteria were fixed using 2.5% glutardialdehyde and were kept in PBS for microscopy. A confocal laser-scanning microscope (CLSM, Leica TCS SP8, Leica Microsystems) was used with a 488 nm excitation laser line and an emission detection at 500–550 nm for Syto^®^9 and a 552 nm excitation laser line and an emission detection at 650–750 nm for propidium iodide. For each sample, 6 images with an area of 190 x 190 µm^2^ were taken at different positions on the glass ring. From the resulting images, the percentage of bacteria with intact (Syto^®^9) and damaged (propidium iodide) membrane was calculated using the ImageJ software 1.48v (Wayne Rasband, National Institute of Health, Bethesda, MD, USA, http://imagej.nih.gov/ij/ (accessed on 15 July 2014)).

### 2.6. BacTiter-GloTM Assay

Bacterial metabolic activity was quantified by means of ATP measurement using the BacTiter-GloTM Microbial Cell Viability Assay (Promega Corporation, Mannheim, Germany) according to the manufacturer’s instructions. Bacteria were prepared as described above, but the optical density at 600 nm was adjusted to 0.05 to fit to the assay’s sensitivity range. Bacterial solutions were measured after 5 min incubation at ambient conditions (to resemble time points before and after SCFS). ATP-dependent luminescence was measured using a multi-mode reader (Infinite 200 Pro, Tecan Group Ltd., Männedorf, Switzerland).

### 2.7. Statistical Analysis

In addition to SCFS, all experiments were performed in three biological replicates (different pre-cultures) and three technical replicates to achieve *N* = 9. Data visualization and statistical analysis were conducted using GraphPad Prism software 8.4 (GraphPad Prism Software Inc., La Jolla, CA, USA). Gaussian distribution was assessed using D’Agostion and Pearson omnibus normality test. Significant differences were analyzed using Wilcoxon test for non-parametric paired data as well as Mann–Whitney U-test for non-parametric and unpaired t-test for parametric unpaired data sets as specified for the respective results. For tests comparing two parameters, significant differences were analyzed using repeated measures two-way ANOVA with Sidak’s multiple comparison correction. Statistical significance was assessed at *p* ≤ 0.05 for all analyses and is referred to as “significant” in the Results and Discussion sections. For linear regression, first, all parameters were transcribed into relative values reflecting the species-specific differences between results in PBS to RTF buffer. These relative differences were then plotted for each two parameters for all species and simple linear regression was performed. Additionally, simple linear regression was also performed excluding the values of *S. oralis*. The goodness of fit is given as R2.

## 3. Results

### 3.1. Titanium Specimen Characteristics

The titanium specimens used as test surfaces for this study exhibited uniform topographies (Figure 2A) with determined roughness values of: arithmetic mean roughness Ra = 0.3 ± 0.05 µm, average surface roughness Rz = 2.6 ± 0.1 µm and maximum roughness depth Rmax = 3.3 ± 0.2 µm. Water contact angle measurement showed a contact angle of 70° ± 14°. The specimen’s zeta potential in both PBS and RTF buffer was determined by streaming potential measurement. As shown in Figure 2B, the zeta potential gradually decreased with increasing pH in both buffers. Whereas at very low pH, zeta potential is slightly higher in PBS, at the pH used for all bacterial experiments (pH 7.5 and pH 7.0 for PBS and RTF, respectively), zeta potential is approx. −50 mV in both buffers.

### 3.2. Strain- and Buffer-Dependent Adhesion Forces of Oral Bacteria to Titanium 

Adhesion forces of six different oral strains to the titanium test surfaces were measured by SCFS using the FluidFM technology. The resulting force–distance curves contained one major and several minor adhesion peaks, as shown in Figure 1 and Figure 3A. The major adhesion peak mostly close to the surface was quantified as maximum adhesion force (Figure 3B) and the minor peaks at different distances were quantified as attachment points (Figure 3C). The distance until the curve returns to the baseline was quantified as detachment distance (Figure 3D). Adhesion forces were measured in PBS and RTF buffer and mean and standard deviation of all values are given in Table 2. The parameters quantified from the force–distance curves are within a common range for all species, except for maximum adhesion force of *S. oralis* in PBS, which is almost four-fold higher. With only few exceptions, adhesion parameters for all species appeared to be buffer specific, mainly with statistically significant lower values in RTF buffer.

### 3.3. Strain- and Buffer-Dependent Biophysical Cellular Characteristics

As a basis for the correlation of adhesion forces to cellular characteristics, bacterial cell elasticity, cellular surface charge, membrane integrity and metabolic activity were quantified for the six strains in PBS and RTF. All results are depicted in Figure 4 and mean values are given in Table 2. Bacterial cell elasticity (Figure 4A) could be determined as Young’s modulus from the SCFS approach curves. Exemplary curves are shown in Appendix A. The steeper the curves are, the higher the Young’s modulus and, thus, the more rigidity is subtended by the bacterial cell. Values appeared to be species specific with, in most cases, statistically significant influences by the different buffers. Bacterial cell’s net surface charge was measured as zeta potential (Figure 4B). All bacteria exhibited negative net surface charge at the given pH with statistically significant reduced zeta potentials in RTF buffer for most species. To determine the distribution of bacteria with intact or damaged membrane (Figure 4C) in the experimental setup of FluidFM-based SCFS, sedimented cells were fluorescently stained and evaluated by CLSM. In contrast to the other cellular parameters, except for *A. ac*, live/dead distribution was similar for all species. It also did not change depending on the buffer. Finally, bacterial metabolic activity after the duration of an SCFS measurement session (5 h) was determined by ATP quantification (Figure 4D). Here, values were highly species specific, not only for the amount of metabolic activity itself, but also regarding buffer dependency, which spanned from statistically significant decrease to no influence and statistically significant increase.

### 3.4. Distinct Correlation between Adhesion Force Parameters and Biophysical Cellular Characteristics

To draw a conclusion for whether the different adhesion force parameters (maximum adhesion force, attachment points, detachment distance) depend on cellular characteristics on a general level, relative changes between values in PBS and RTF were plotted for each strain and linear regression was performed. The results are given in Figure 5. If the values for *S. oralis* were included (gray dots), no correlation could be detected, as also indicated by low correlation coefficients (gray values). However, if *S. oralis* was excluded from analysis (only black dots and values), two distinct types of correlations, each with R2 > 0.9, could be identified. On the one hand, maximum adhesion forces correlated positively with cellular elasticity and negatively with the absolute value of bacterial surface charge measured as zeta potential. As shown in Appendix A, also, cellular elasticity and bacterial surface charge correlated directly with R2 = 0.75, whereas all other cellular parameters did not. On the other hand, number of attachment points and detachment distance correlated positively with metabolic activity. Interestingly, even though correlation with metabolic activity did not apply for *S. oralis*, the direct correlation of number of attachment points with detachment distance could be detected across all species analyzed (Appendix A). In contrast, these parameters did not correlate to maximum adhesion force (Appendix A).

## 4. Discussion

Bacterial adhesion to dental implants is the onset for the development of pathological biofilms. Therefore, characterization of this initial step is the basis towards the development of novel biofilm-preventive implant materials. In this way, the present study aimed to comparatively measure adhesion forces of oral bacteria of different ecological niches and evaluate the general influence of cellular characteristics on these results.

Bacterial adhesion was analyzed on titanium, one of the most common implant materials. Roughness of the test specimens was chosen to match established parameters for reliable control surfaces routinely used in antibacterial materials research [32,33,34,35,36,37,38,39]. The surface roughness sometimes caused scattering photons in the AFM laser to interfere with the reflected laser beam, resulting in incomplete approach or sinus-shaped artefacts in the force–distance curves. To take this into account, all force–distance curves were manually controlled before analysis. At ambient conditions, titanium surfaces carry a titanium dioxide layer and the protonation or deprotonation of hydroxyl groups on this oxide layer is responsible for their surface charge [40]. The zeta potential measured here is directly related to this surface charge in PBS and RTF. At very low pH, most hydroxyl groups are protonated and surface charge is less negative, especially when further supported by a high ionic strength in PBS. With increasing pH, the proportion of deprotonated hydroxyl groups increases, which results in a similarly decreased zeta potential in both buffers. The negative titanium surface charge at physiological pH values is in line with other studies [41,42]. The two buffers served as simplified models for different environmental conditions for the following experiments. PBS is a standard salt solution using phosphate as a buffering agent that is often used for cell and bacteria culture in medical research, as it is isotonic to the human body. RTF is a more complex balanced mineral salt solution that contains also dithiothreitol to make it oxidation resistant. It was specifically designed as a storage medium for (facultative) anaerobic oral bacteria [24]. In addition to chemical composition and oxygenation, PBS and RTF also differ in their ionic strength, which is approx. 60% lower for RTF. As zeta potential was similar for both buffers at physiological pH, similar material properties for the following experiments could be ensured and do not have be taken into account when correlating adhesion forces and cellular characteristics.

To directly measure bacterial adhesion forces to titanium surfaces on a single-cell level, a FluidFM (AFM connected to a microfluidic pressure control system)-based system was used. The bacterial adhesion time within this study was set to five seconds. Previous studies have shown that, already, directly after surface contact, initial adhesion forces can be measured (adhesion time of zero seconds) [11,32,43,44]. Increased adhesion times would cause removal of interfacial water and subsequent bond strengthening with enhanced adhesion forces [11,32,43]. Another important parameter for SCFS that positively correlates with adhesion forces is the setpoint force, which is used to approach the bacterium to the surface [11]. Here, setpoint force of 0.75 nN was used to avoid bacterial compression but ensure a reliable approach [45]. As a first step towards the characterization of different oral bacteria adhesion forces to the implant material titanium, measurements were taken for six bacterial strains of the diverse oral microbiome (*S. oralis*, *A. naeslundii*, *V. dispar*, *P. gingivalis*, as well as rough and smooth *A. ac*) at this fixed setpoint force. From the resulting force–distance curves, different parameters were quantified: maximum adhesion force, which is driven by unspecific Lifshitz–Van-der-Waals and electrostatic forces [43,46,47,48], the number of single attachment points, which are specific interactions of bacterial surface molecules [43,46,47,48] and the detachment distance that reflects the length as well as the stretching capability of these surface molecules and to a minor portion stretching of the bacterial cell [45,47]. 

In the literature, adhesion force studies of several oral bacteria, mainly streptococci, have already been conducted [16,17,32,49,50,51,52]. Amongst these, *S. oralis* adhesion has been analyzed on several dentistry-related materials. Resulting adhesion forces were in a range of 0.25–4 nN. This is a broader spectrum but is in agreement with the data obtained in this study, even though setpoint forces and surfaces varied and, in most studies, bacteria-coated AFM cantilevers were used. The number of attachment points and the detachment distances were not quantified in all studies. Values that can be found in the literature are two–six attachment points and detachment distances of 0.2–0.8 nm for streptococci [16,32,43,44]. Whereas the highly surface-dependent attachment points clearly differ, the detachment distance is comparable to the results of this study, as it is only to a minority influenced by a rigid, non-elastic surface. *A. naeslundii* was also analyzed for adhesion forces on dentistry-related materials and to coaggregation partners using bacteria-coated cantilevers [16,52,53,54]. The measured adhesion forces of 0.2–6.1 nN are higher than those in this study, which can be mainly attributed to the different experimental conditions used. Detachment distances that were reported for *A. naeslundii* are comparable to those in this study [52]. For rough *A. ac*, whose fimbriae contain Flp proteins as the main structural component, the isolated protein was already subjected to adhesion force measurements by directly coating cantilevers [55]. The results greatly vary from “not measurable” up to 10 nN, depending on the surface and are, thus, also in the range of values detected here. *P. gingivalis* adhesion forces to titanium were measured by FluidFM in a recent material-focused study, where only the maximum adhesion force was quantified with approx. 2.5 nN [51]. This is higher than the values of this study, which is most probably due to the unmachined titanium surface with large valleys in their study. To the best of our knowledge, for rough and smooth *A. ac*, but also *V. dispar*, no single-cell adhesion force experiments have been performed before. 

As, in the present study, a similar setup was used for all bacterial species, it allowed for a direct comparison of their adhesion forces with regard to their role in oral biofilm development. *S. oralis* and *A. naeslundii* showed the highest values for maximum adhesion force and attachment points, especially in physiological PBS buffer, followed by *P. gingivalis* and *A. ac*. Comparably low adhesion forces were detected for *V. dispar*. *S. oralis* and *A. naeslundii* are oral pioneer bacteria that are among the first to adhere on solid substrates in the oral cavity. Comparable higher adhesion forces of commensal pioneer bacteria, such as mitis group Streptococci, have, likewise, been described in the literature [16,17]. This correlates to their dominance in early biofilms. In contrast, *P. gingivalis* and A.ac are oral pathogens that contribute to oral biofilms at a later state. In pathogenic species, adhesion abilities are regarded as virulence factors. As this adhesion mainly focusses on human tissue and other bacteria, their adhesion to solid surfaces may be reduced compared to initial colonizers. *V. dispar* is a non-virulent, commensal secondary colonizer that coaggregates with *S. oralis* and *A. naeslundii* [27]. Thus, there might be no evolutionary pressure towards the development of strong adhesion forces to solid surfaces. Within its limitations, the results of this study showed interesting similarities between adhesion forces of bacteria measured by SCFS and their role in oral biofilms and encourage further studies towards this direction.

Additionally, for *A. ac*, a rough and a smooth colony forming strain was analyzed. Rough *A. ac* is a virulent wild-type strain that exhibits fimbriae, which mainly consist of Flp proteins [55]. Smooth *A. ac* is a mutant that lacks these fimbriae, which is mainly attributed to mutations in the *flp* promotor region [56]. This difference is considered to be the main reason for an increased surface adhesion and virulence of rough *A. ac* [55,57]. Interestingly, in the present study, no increased adhesion forces could be detected for rough *A. ac* compared to the smooth strain. Adhesion forces of smooth *A. ac* were even slightly but significantly higher in RTF buffer. It has already been shown that adhesion proteins also contribute to *A. ac* adhesion and enable biofilm formation of the smooth strain [25,29]. Additionally, studies demonstrated that initial *A. ac* adhesion is mainly driven by unspecific electrostatic forces that are independent from specific protein–surface interactions [55]. In contrast to previous studies, where comparatively long-term adhesion was analyzed, in the present study, the adhesion time was only five seconds. Thus, it can be assumed that mainly electrostatic, not protein-specific forces, contributed to the adhesion forces measured here, which reduced the effect of the lacking fimbriae. The slight differences that still could be observed confirm the different cell surface composition of rough and smooth *A. ac*.

Bacterial adhesion forces were lower in RTF compared to PBS buffer for most of the species analyzed. This could be most probably attributed to differences in ionic strength. It has already been reported that several bacterial strains show positive correlation of adhesion forces and ionic strength [42,58]. However, for other strains, negative correlation was demonstrated [59]. This points towards a certain strain dependency and possible further factors that might be involved. It is well established that biophysical cellular characteristics influence the bacteria–surface interaction [19,22,23]. On the way to a more detailed understanding of the adhesion of different oral bacteria to implant surfaces, it is, thus, necessary to know how these parameters influence bacterial adhesion forces on a general level. Therefore, cell elasticity, cellular surface charge, membrane integrity and metabolic activity were quantified in both buffers in a setup similar to SCFS. In a second step, the relative differences between values in PBS and RTF were calculated for each strain and parameter and linear regression analysis was performed between adhesion force values and cellular characteristics.

Bacterial single-cell elasticity was quantified by pressing onto the bacterial cells during SCFS approach. To the best of our knowledge, single-cell elasticity has not been quantified before for the species in this study. It was shown that Young’s moduli of other bacterial species vary greatly and range from approx. 100 kPa to 200 MPa [19,60]. The difference to the values detected in this study, which were 12–180 kPa, may be attributed to different AFM settings and environmental conditions. In the present study, bacterial elasticity changed in a species-specific manner according to the buffer used. In contrast to some hypotheses, there is no evidence that Gram-negative cells (*V. dispar*, *P. gingivalis*, *A. ac*) are, in general, more elastic than Gram-positive cells (*S. oralis*, *A. naeslundii*), which supports the notion that there is no simple correlation between both factors [19]. When relating single-cell elasticity to the parameters of adhesion force measurement, there is no general correlation at first glance. However, when excluding the values for *S. oralis*, a strong positive correlation between maximum adhesion force and cell elasticity can be observed for all other species. It can be presumed that for these cells, an increased flexibility enables stronger membrane–surface contact and, thus, increases electrostatic interaction. This is in line with the already described viscoelastic behavior of bacteria during adhesion and the importance of bacterial elasticity for adhesion to nanostructured surfaces [61,62,63]. The different behavior of *S. oralis* requires further studies. One hypothesis might be that amongst the bacteria analyzed, it is the only Gram-positive strain that attaches to surfaces directly by membrane-bound adhesins (Table 1). *A. naeslundii* is also a Gram-positive bacterium but has additional fimbriae as appendices (Table 1), whereas all other species are Gram negative. Thus, future experiments should re-analyze the correlation between cell elasticity and adhesion forces with specific regard to cell wall structures.

The charge of bacterial surfaces was quantified as zeta potential. It reflects the net electrical charge of surface molecules and, thus, is the average charge of membrane phospholipids, functional groups (such as peptidoglycans, teichonic acids or lipopolysaccharides) and surface proteins in their three-dimensional structure [21,64,65]. As these molecules are mostly negatively charged at physiological pH, bacterial zeta potential was negative for all species analyzed. As for cell elasticity, there is no evidence that the different surface compositions of Gram-positive and Gram-negative bacteria influence surface charge on a general level. Instead, it seemed to depend on the specific surface composition of each strain [21]. As charge develops as a result of protonation and deprotonation of surface molecules, it depends on environmental conditions, such as pH or ionic strength [21,66]. It has already been shown that bacterial zeta potential is less negative at higher ionic strength [42,58], most probably due to the higher availability of ions counterbalancing the surface charge. In line with this, for most bacterial species analyzed in this study, zeta potential was lower (more negative) in RTF than in PBS. When correlating bacterial surface charge to results of adhesion force measurements, first, it has to be mentioned that the influence of titanium surface charge on this interaction can be neglected as zeta potential is similar in both buffers at the given pH values. For all species except *S. oralis,* a strong negative correlation between both factors could be detected. This is supported by the literature and most probably due to the fact that a stronger negative charge of bacterial surfaces causes greater repulsion from the negative-charged titanium surface [19,42,67]. As there is also a (less strong) correlation between zeta potential and cell elasticity for all species except *S. oralis*, it might additionally be assumed that differences in cell elasticity in the different buffers are also due to different charges and, thus, interactions of surface molecules. On the other hand, there are also studies that did not find a correlation between surface charge and bacterial attachment [19]. Taking into account the different results for *S. oralis*, this further supports the need for a more detailed study of bacterial adhesion dependency on different cell wall structures.

Bacterial membrane integrity was determined by live/dead fluorescent staining. It is based on different membrane permeability of the dyes Syto^®^9 (permeable) and propidium iodide (non-permeable) and mostly used to quantify viability rates of bacterial cells (live/dead ratio), as impaired membranes are often defined as dead cells [33,68]. A decrease in membrane integrity could, thus, be equated to a decreased viability. Even though values for membrane integrity varied between species, no species showed differences in membrane integrity between PBS and RTF. As both buffers are intended to maintain bacterial viability, these results could be expected. Consequently, there is also no correlation between membrane integrity and the parameters of adhesion force measurement. Probably, as membrane integrity also does not correlate to any other cellular parameter, the increase in permeability that allows propidium iodide to enter the cell does not immediately alter cellular surface molecules and surface charge. The process of initial adhesion might, thus, not be directly affected. 

The last cellular parameter analyzed was bacterial metabolic activity on the basis of ATP amount. As this molecule is only produced in metabolic-active cells, not stored and rapidly degraded upon cell death, it directly correlates to bacterial metabolism [69,70]. For the setup of this study, bacterial metabolic activity appeared to be highly species and also buffer dependent. As the bacteria’s respiratory chain creates a gradient on the surface of the cell membrane, the environmental conditions directly influence the respiratory efficacy. The metabolic activity, thus, also reflects how well the buffer’s ionic strength, pH and oxygenation suit the requirements for sufficient cellular respiration for the different species. When correlating these results to the parameters of adhesion force measurement, a strong positive correlation to the number of attachment points and the detachment distance could be detected—again, when excluding values for *S. oralis*. Additionally, the number of attachment points and the detachment distance strongly correlate themselves across all species analyzed. As the former represents specific surface molecule interactions and the latter stretching of surface molecules [43,46,47,48], this correlation had to be expected. It can be hypothesized that for the species other than *S. oralis,* an increased metabolic activity results in an increase in surface-adhesion molecules. These can establish more attachment points and also allow for more molecule stretching, which, in turn, increase the detachment distance. The latter holds for *S. oralis* as well; however, it is not linked to an increase in surface molecules upon increased metabolic activity. This might again be due to the different cell surface composition of this Gram-positive, non-fimbriated species.

## 5. Conclusions

The results of this study present a step towards the characterization of oral bacteria’s adhesion to titanium surfaces. Adhesion forces of different species in the diverse oral biofilm community could reliably be measured within the same experimental setup. This also allowed for direct comparison with regard to their role in biofilm development, which indicated the strongest adhesion forces for pioneer commensals and lowest for co-aggregation specialized secondary colonizers. From this first step towards a better understanding of bacterial adhesion in oral biofilm formation, future SCFS studies should dive deeper into this topic by, e.g., analyzing force development over time and on different materials, also including saliva-conditioning films, comparison of bacteria–surface vs. bacteria–bacteria adhesion forces, differences in adhesion forces between different strains of the same species as well as adhesion forces to human cells.

The second focus of this study was the correlation of adhesion force results with biophysical cellular characteristics across the species analyzed. Here, distinct correlations between electrostatically driven maximum adhesion force, bacterial surface elasticity and surface charge as well as single-molecule attachment points, stretching capability and metabolic activity could be identified. The different results for Gram-positive, non-fimbriated *S. oralis* encourages further studies that re-analyze these correlations with regard to bacterial cell surface composition. Unraveling the underlying mechanisms could finally be used for knowledge-driven development of novel antiadhesive implant materials.

## Figures and Tables

**Figure 1 bioengineering-09-00567-f001:**
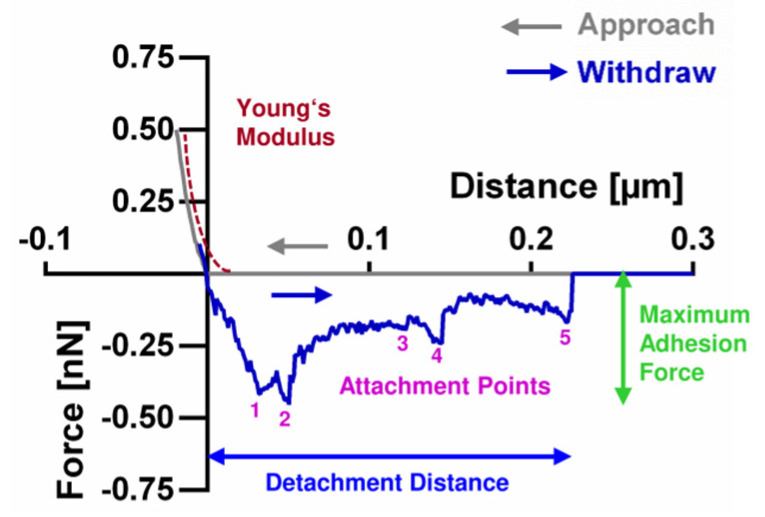
Schematic illustration of parameters quantified from force–distance curves of single-cell force spectroscopy. The grey curve shows the approach of a bacterium to the surface, whereas the blue curve shows its subsequent withdrawal.

**Figure 2 bioengineering-09-00567-f002:**
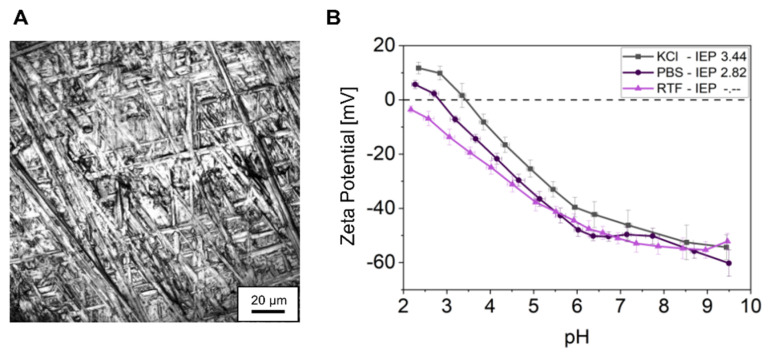
(**A**) Representative reflection microscopy image of the titanium surface. (**B**) Zeta potentials of titanium surfaces mean ± SD in PBS and RTF buffer as well as in KCl reference as a function of different pH; isoelectric points (IEPs) are given in the inset.

**Figure 3 bioengineering-09-00567-f003:**
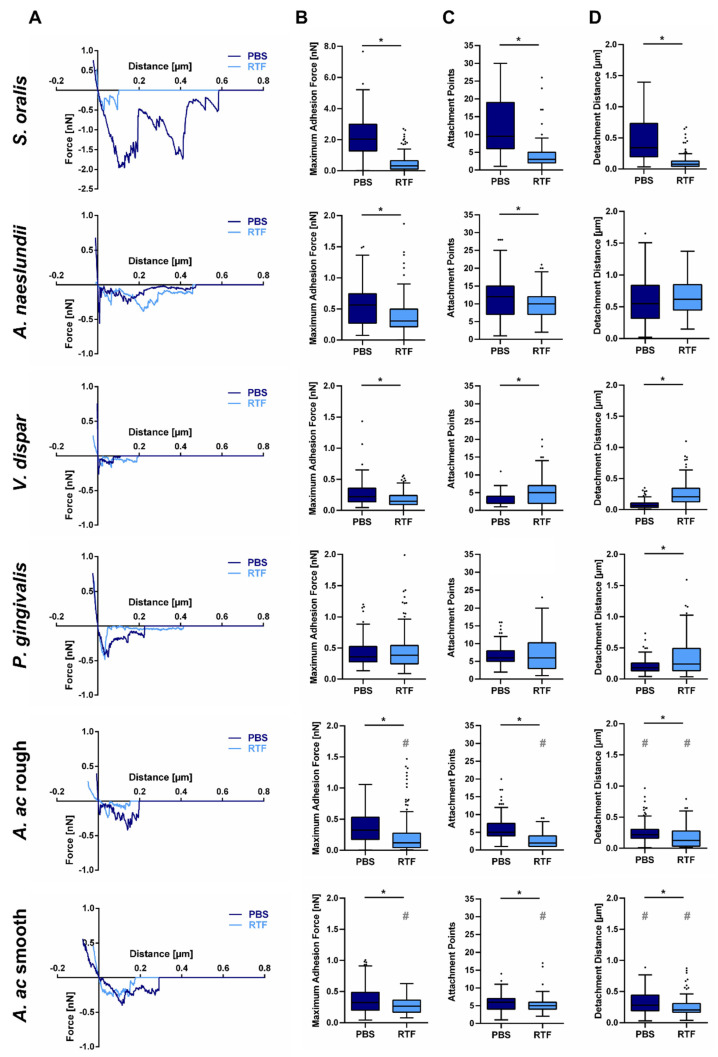
Single-cell adhesion forces of oral bacteria on titanium in different buffers. (**A**) Exemplary force–distance curves and Tukey boxplots of (**B**) the maximum adhesion force, (**C**) the number of attachment points and (**D**) the detachment distance of indicated strains in PBS and RTF buffer measured by SCFS with 5 s adhesion time and setpoint force of 0.75 nN. * indicates significant differences for each strain and # indicates significant differences between rough and smooth *A. ac* in the corresponding buffer with *p* ≤ 0.05 according to Mann–Whitney U-test.

**Figure 4 bioengineering-09-00567-f004:**
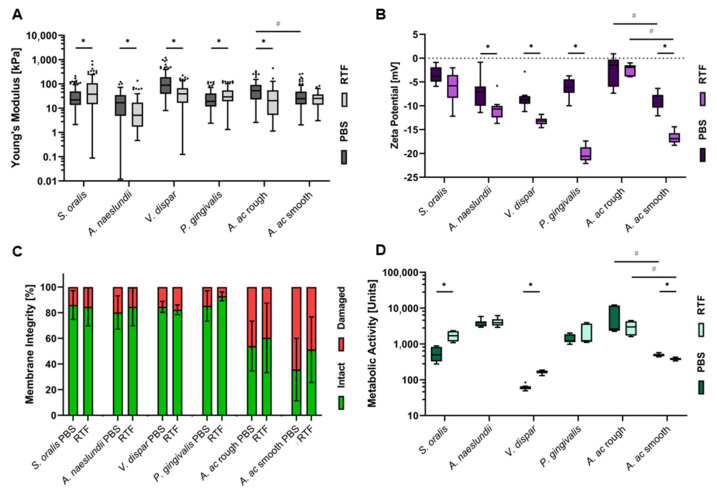
Cellular characteristics of oral bacteria in different buffers. (**A**) Tukey boxplots of bacterial Young’s moduli measured by SCFS with setpoint force of 0.75 nN, (**B**) Tukey box plots of bacterial zeta potential, (**C**) percentage in mean ± standard deviation of bacteria with intact and damaged membrane detected by fluorescence staining and CLSM of indicated strains after 5 h incubation and (**D**) Tukey boxplots of bacterial metabolic activity after 5 h incubation measured by ATP quantification of indicated strains in PBS and RTF buffer. * indicates statistically significant differences for each strain and # indicates statistically significant differences between rough and smooth *A. ac* in the corresponding buffer with *p* ≤ 0.05 according to Wilcoxon test (**A**), Mann–Whitney-U test (*A. naeslundii* in (**B**), all strains in (**C**)) and unpaired t-test (all other strains in (**B**), all strains in (**D**)).

**Figure 5 bioengineering-09-00567-f005:**
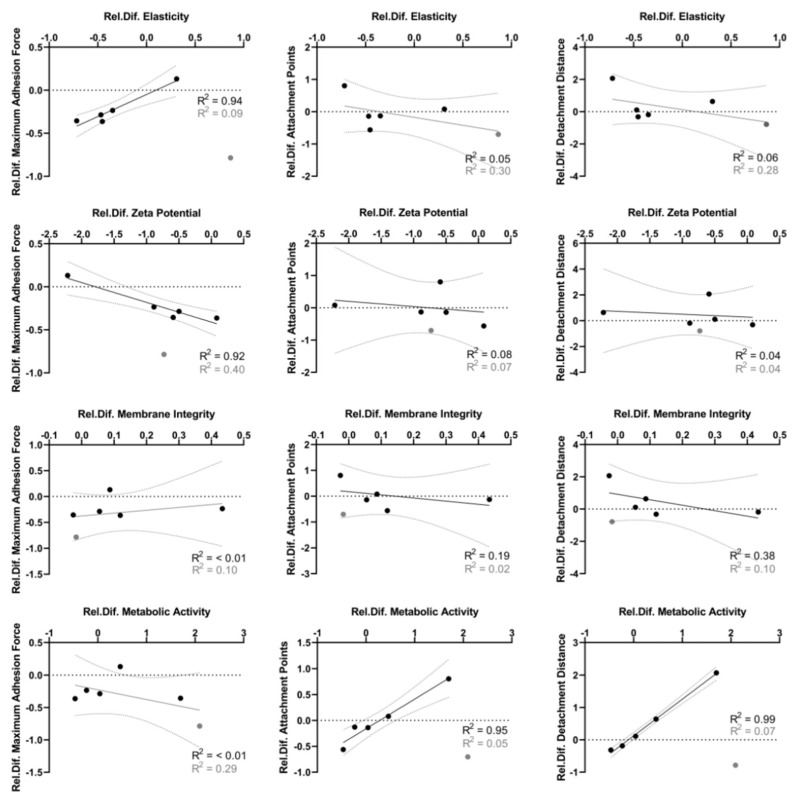
Linear regression analysis between cellular characteristics and adhesion force measurement results across species. Each black dot represents the relative difference of respective absolute values in PBS to RTF buffer for one species. Values for *S. oralis* are shown as gray dots. Linear regression fit is shown with 95% confidence interval and the corresponding correlation coefficient R2. For black lines and R2, *S. oralis* was excluded from calculation. Gray lines and R2 also include the gray values of *S. oralis*.

**Table 1 bioengineering-09-00567-t001:** Characteristics of bacterial strains analyzed in the study.

Species	Shape	Size [µm]	Metabolism	Cell Wall	Adhesion Structures	Characteristic in Oral Biofilm
*S. oralis*	coccoid	0.75 × 0.75	facultative anaerobe	Gram+	fibrils adhesins [26]	commensal bacterium initial colonizer
*A. naeslundii*	bacillus	3 × 0.75	facultative anaerobe	Gram+	fimbriae [27]	commensal bacterium initial colonizer
*V. dispar*	coccoid	1.2 × 1.2	anaerobe	Gram−	fimbriae adhesins [27]	commensal bacterium secondary colonizer
*P. gingivalis*	coccobacillus	1 × 0.75	anaerobe	Gram−	fimbriae adhesins [28]	oral pathogen late colonizer
*A. ac* rough	coccobacillus	1 × 0.75	facultative anaerobe	Gram−	fimbriae adhesins [29]	oral pathogen late colonizer
*A. ac* smooth	coccobacillus	1 × 0.75	facultative anaerobe	Gram−	adhesins [29]	oral pathogen late colonizer

**Table 2 bioengineering-09-00567-t002:** Summary of parameters analyzed for bacterial strains in different buffer conditions as mean ± standard deviation.

Species	Buffer	Maximum Adhesion Force [nN]	Attach. Points	Detach. Distance [µm]	Elasticity [kPa]	Zeta Potential [mV]	Relative Membrane Integrity [%]	Metabolic Activity [Units]
*S. oralis*	PBS	2.19 ± 1.34	13 ± 8	0.46 ± 0.31	38.5 ± 39.9	−3.6 ± 1.8	86.0 ± 11.2	551 ± 238
	RTF	0.47 ± 0.51	4 ± 3	0.10 ± 0.10	71.9 ± 86.1	−6.2 ± 3.2	84.5 ± 14.8	1705 ± 502
*A. naeslundii*	PBS	0.56 ± 0.31	12 ± 6	0.60 ± 0.36	24.3 ± 25.4	−7.2 ± 3.2	80.2 ± 13.0	3936 ± 885
	RTF	0.40 ± 0.28	10 ± 4	0.66 ± 0.30	12.9 ± 18.9	−10.8 ± 2.3	84.5 ± 14.7	4102 ± 1121
*V. dispar*	PBS	0.28 ± 0.20	3 ± 2	0.08 ± 0.07	178.2 ± 234.3	−8.4 ± 2.3	84.5 ± 4.2	61 ± 11
	RTF	0.18 ± 0.12	5 ± 4	0.26 ± 0.19	50.4 ± 42.4	−13.3 ± 0.8	82.3 ± 3.8	165 ± 18
*P. gingivalis*	PBS	0.45 ± 0.33	7 ± 3	0.21 ± 0.12	31.9 ± 27.3	−6.2 ± 2.0	85.3 ± 11.8	1404 ± 394
	RTF	0.51 ± 0.44	7 ± 6	0.34 ± 0.30	41.8 ± 29.6	−20.0 ± 1.6	92.7 ± 3.4	2052 ± 1303
*A. ac* rough	PBS	0.37 ± 0.23	6 ± 4	0.26 ± 0.16	68.3 ± 64.6	−2.6 ± 3.1	54.0 ± 19.5	5632 ± 4735
	RTF	0.24 ± 0.30	3 ± 2	0.18 ± 0.17	37.3 ± 52.3	−2.4 ± 1.2	60.4 ± 27.1	2996 ± 1178
*A. ac* smooth	PBS	0.37 ± 0.23	6 ± 2	0.32 ± 0.17	41.0 ± 47.7	−8.9 ± 1.8	35.7 ± 24.2	496 ± 44
	RTF	0.28 ± 0.13	5 ± 2	0.26 ± 0.16	26.8 ± 16.7	−16.7 ± 1.3	51.2 ± 25.6	380 ± 31

## Data Availability

The data are available upon request from the research data management of Hannover Medical School.

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
