# Peer review of "Adhesion Forces of Oral Bacteria to Titanium and the Correlation with Biophysical Cellular Characteristics"

_bioengineering, 2022, doi:10.3390/bioengineering9100567_

Round 1

Reviewer 1 Report

In my opinion the novellety is missing. There are already a lot of papers dealing with adhesion forces of bacteria on surfaces under the influence of different parameters. In my opinion, here are investigated some types of bacteria which were not investigated yet. So, a real novellety is missing in my opinion.
In addition, I expected a high significance of conente by the title, especially the part "Influence of celluar charactersitics". But the cellular characterisitics were simple (zeta potential, hydrophobicity,...) and I expected more by the title. Another point is, that the conclusion first refers a lot to the use of the Fluid-FM for SCFS. However, the title and the abstract let the reader expect other details/conclusions. In addition, the Fluid FM is already described several times for bacterial adhesion in other literature. Here, in my opinion, literature is missing in the reference list.

Concerning the hydrophobicity the distionction between hydrophilic and hydrophobic should be extended according to Berg et el. According to Berg et al the disctionction has to be done at a contact angle of 65° if biological molecules/organisms interact with solid surfaces.

In addition I am missing an explanation why the zeta potential in RTF is reduced. What could be the reason an what does this reason mean for the bacterial cell characterisitcs?

In the results part some comparison of the adhesion data with literature data is done. However, the informtaion wheter these literatuer data are measured by Fluid FM or by chemically modified tipless cantilevers is missing. And I think the use of Fluid FM respectilvey tipless cantilevers with chemical fixation of the cells can differ a lot the obtained results.

In addition there is a mismatch between the bacterial zeta potential in the materials and methods part and the results part. In materials and methods, there is written it was used the diffuson barrier method. In the results part, there was used the word microelectrophoresis. This can be confusing. Please provide any information concering microelectrophoreisis in the materials and methods part.

Author Response

We would like to thank the reviewer for the detailed analysis of our manuscript and helpfull comments, which helped to further sharpen and improve it. Please find in the following a detalied answer to all concerns:

In my opinion the novellety is missing. There are already a lot of papers dealing with adhesion forces of bacteria on surfaces under the influence of different parameters. In my opinion, here are investigated some types of bacteria which were not investigated yet. So, a real novellety is missing in my opinion.

The reviewer is right. Oral bacteria’s adhesion forces have already been measured. Also, the influence of surface charge, elasticity, etc. has been analyzed before. However, in our opinion the novelty of our study lies in the combination. To the best of our knowledge, so far, no study analyzed the adhesion forces of several species of different niches of oral biofilms in one study. Other studies mostly concentrated on viridans streptococci and actinomyces, which are both primary colonizers. As the results of SCFS greatly depend on the specific measuring setups, analyzing the different species within one experiment allows for a direct comparison and can, thus, provide novel insights into the oral microbial community. Likewise for the analyzed cellular characteristics: they have been analyzed individually, but not systematically for all of these factors across different species within one experimental setup so far. We modified the abstract and the introduction (line 62 – 89) to point this out more strongly.

In addition, I expected a high significance of conente by the title, especially the part "Influence of celluar charactersitics". But the cellular characterisitics were simple (zeta potential, hydrophobicity,...) and I expected more by the title. Another point is, that the conclusion first refers a lot to the use of the Fluid-FM for SCFS. However, the title and the abstract let the reader expect other details/conclusions.

The authors apologize, if title and abstract misled the reviewer. The part of “influence of cellular characteristics” was changed to be more precisely. To be correct, we did not analyze the influence in general, but statistical correlations. Also, the cellular characteristics are limited to some biophysical ones. Furthermore, the correlation was drawn across the six species of this study, not on a common general level. These specifications were changed in the title, the abstract, the conclusion and in the wording throughout the manuscript.

In addition, the Fluid FM is already described several times for bacterial adhesion in other literature. Here, in my opinion, literature is missing in the reference list.

Within the adaptation of the manuscript, as described above, literature was updated and completed as well. As this caused changes throughout the manuscript and would include formatting comments at each citation, this was not made in “track changes” mode for the sake of readability.

Concerning the hydrophobicity the distionction between hydrophilic and hydrophobic should be extended according to Berg et el. According to Berg et al the disctionction has to be done at a contact angle of 65° if biological molecules/organisms interact with solid surfaces.

The reviewer is right. To precisely determine, if a surface exhibits hydrophilic or hydrophobic properties, more information than only the water contact angle is necessary. E.g., for interpretation based on DIN ISO, analysis of two different substances would be required. Therefore, the labeling as “hydrophilic” of our titanium surfaces was removed from the text (line 241).

In addition I am missing an explanation why the zeta potential in RTF is reduced. What could be the reason an what does this reason mean for the bacterial cell characterisitcs?

Regarding zeta potential of titanium surfaces, the slightly lower values at very low pH are most probably due to an increased protonation of local hydroxyl groups in buffer with higher ionic strength (here: PBS). The surface charge gradually decreases with increasing pH, as the hydroxyl groups get deprotonated. At physiological pH, as used for SCFS experiments, the surface charge in both buffers was similar and could, thus, be neglected when correlating adhesion forces and cellular characteristics. A more detailed explanation was added from line 354 to 359.

Regarding bacterial zeta potential, it has already been shown that less negative values are achieved in conditions with higher ionic strength, most probably due to a higher availability of ions counterbalancing any surface charge. When correlating this to bacterial adhesion forces (excluding S. oralis), a strong negative correlation could be observed, which is most probably due to a stronger repulsion from a likewise negative charged titanium surface. This discussion was already part of the manuscript and is highlighted in line 498ff and 504ff.

In the results part some comparison of the adhesion data with literature data is done. However, the informtaion wheter these literatuer data are measured by Fluid FM or by chemically modified tipless cantilevers is missing. And I think the use of Fluid FM respectilvey tipless cantilevers with chemical fixation of the cells can differ a lot the obtained results.

The information was added in the paragraph starting from line 390.

In addition there is a mismatch between the bacterial zeta potential in the materials and methods part and the results part. In materials and methods, there is written it was used the diffuson barrier method. In the results part, there was used the word microelectrophoresis. This can be confusing. Please provide any information concering microelectrophoreisis in the materials and methods part.

The authors apologize for this mistake. Bacterial zeta potential was determined by diffusion barrier method. Microelectrophoresis was deleted (line 488).

Reviewer 2 Report

The authors investigated a step for adhesion force of oral bacteria to titanium and cellular characteristics were analyzed, which is presented well as experimental design and writing. While there are some comments as below:

1.     For fig 2b, the y axis number of zeta potential is from 0 to -60, should be changed as negative zeta potential. polydispersity index            is an important indicator for validating the zeta potential, the authors need to disclosure it.

2.     For force spectroscopy measurement, did authors maintain the temperature as 370C to mimic in vivo environment?

Author Response

We would like to thank the reviewer for the helpfull comments on our manuscript. Please find in the following a detalied answer to all concerns:

For fig 2b, the y axis number of zeta potential is from 0 to -60, should be changed as negative zeta potential. polydispersity index   is an important indicator for validating the zeta potential, the authors need to disclosure it.

Thank you, for pointing this out. The formatting mistake in fig 2 was reversed. The PDI was not determined during zeta potential measurement, as the bacterial size is known and varies only within a small spectrum. They can be considered as homogenous particles. Also, the zeta potential was determined by diffusion barrier method (the wrongly written microelectrophoresis in the discussion was deleted in line 488). This method avoids aggregation, as the particles are not in direct contact with the electrodes and, thus, changes in surface charge.

For force spectroscopy measurement, did authors maintain the temperature as 370C to mimic in vivo environment?

No, the measurement was done at ambient temperature. The AFM setup exploited for this study does not include an incubation chamber and, thus, measurement at 37°C was not possible. But it would be interesting to address this topic in future investigations, as most SCFS studies so far have been performed at ambient temperature.